# Multisensory integration in the developing tectum is constrained by the balance of excitation and inhibition

**Daniel L Felch[1,2], Arseny S Khakhalin[1,3], Carlos D Aizenman[1]\***

[1]Department of Neuroscience, Brown University, Providence, United States; [2]Department of Cell and Molecular Biology, Tulane University, New Orleans, United States; [3]Department of Biology, Bard College, New York, United States

**Abstract** Multisensory integration (MSI) is the process that allows the brain to bind together spatiotemporally congruent inputs from different sensory modalities to produce single salient representations. While the phenomenology of MSI in vertebrate brains is well described, relatively little is known about cellular and synaptic mechanisms underlying this phenomenon. Here we use an isolated brain preparation to describe cellular mechanisms underlying development of MSI between visual and mechanosensory inputs in the optic tectum of Xenopus tadpoles. We find MSI is highly dependent on the temporal interval between crossmodal stimulus pairs. Over a key developmental period, the temporal window for MSI significantly narrows and is selectively tuned to specific interstimulus intervals. These changes in MSI correlate with developmental increases in evoked synaptic inhibition, and inhibitory blockade reverses observed developmental changes in MSI. We propose a model in which development of recurrent inhibition mediates development of temporal aspects of MSI in the tectum.

**\*For correspondence:**
Carlos_Aizenman@brown.edu

**Competing interests:** The authors declare that no competing interests exist.

## Introduction

Multisensory integration (MSI) is a well-characterized phenomenon of both neural output and behavior where the response triggered by a stimulus of a given sensory modality is altered by the coincident presentation of a stimulus of a different sensory modality (for reviews see *Stein and Stanford, 2008*; *Stein et al., 2009*). The result can be either enhancement or suppression of the response relative to the responses evoked by a single unimodal stimulus (*Stanford et al., 2005*).

One central brain area where inputs from multiple sensory modalities first interact is the optic tectum. This midbrain structure, also known as the superior colliculus in mammals, contains laminae that are segregated functionally, as well as anatomically—its superficial layers receive a direct retinal projection that maintains its topographic map of visual space (*Graybiel, 1975*; *Straznicky and Gaze, 1972*) and its deep and intermediate layers receive inputs carrying both auditory information (*Knudsen, 1982*; *Lowe, 1986*) as well as somato- and mechanosensory information and motor feedback signals (for a review see *May, 2006*). In a given multisensory neuron in the colliculus, the amount of integration is dependent on both the overlap between spatial receptive fields for the two stimulus modalities and the time window between stimulus presentations in a crossmodal pair (*Meredith et al., 1987*; *Meredith and Stein, 1996*). The integrative capabilities of multisensory collicular neurons develop in an activity dependent manner which has been proposed to depend on repeated associations of spatiotemporally aligned stimuli during a critical period (*Knudsen, 2002*; *Knudsen and Brainard, 1991*; *Wallace et al., 2006*; *Wallace and Stein, 2007*; *Xu et al., 2012*; *Yu et al., 2010*).

Despite great interest within the field in understanding potential cellular and synaptic mechanisms for multisensory integration and its development (*Rowland et al., 2007b*), very little experimental work exists on this question (*Binns and Salt, 1996*; *Skaliora et al., 2004*). The prevalent experimental models of multisensory integration have been the superior colliculus of cats and rodents and to some degree the optic tectum of the barn owl; their location within the animal makes these structures difficult to access with whole-cell recording techniques in vivo and impossible to isolate in vitro without significantly disrupting the neural networks in which they operate. In Xenopus laevis tadpoles, the optic tectum is superficially located and thus accessible for whole-cell recordings in vivo (*Khakhalin et al., 2014*) and can be isolated in an intact whole-brain preparation for recordings ex vivo (*Pratt and Aizenman, 2007*). This is a tremendous advantage over other preparations, where assessing synaptic events during MSI is usually inferred from local extracellular field potentials.

In Xenopus, as in other vertebrates, the dominant input to the optic tectum is visual, originating from the contralateral retina, and terminating in the superficial layers of the tectal neuropil (*Deeg et al., 2009*; *Székely and Lázár, 1976*). Inputs from the various mechanosensory modalities that ascend via hindbrain nuclei terminate within the deeper layers (*Behrend et al., 2006*; *Deeg et al., 2009*; *Hiramoto and Cline, 2009*; *Lowe, 1986*; *1987*). Our laboratory and others have characterized the development of hindbrain mechanosensory projections to the tectum in Xenopus tadpoles, and have shown that these inputs converge in single tectal neurons from very early developmental stages. In this study we use the Xenopus tadpole to begin to examine the cellular and circuit level mechanisms underlying multisensory integration of these hindbrain and visual inputs in the developing midbrain.

## Results

### Temporal interactions

Experiments were performed between developmental stages 44–46 and 48–9. During this developmental time span, tectal neurons already have been innervated by both visual and mechanosensory inputs and respond synaptically to activation of either pathway (*Deeg et al., 2009*; *Hiramoto and Cline, 2009*). During this timeframe tectal circuits also undergo a period of refinement, both of sensory receptive fields (*Dong and Aizenman, 2012*; *Dong et al., 2009*; *Tao and Poo, 2005*) and of local recurrent circuitry (*Pratt et al., 2008*). Tectal neurons also undergo dramatic change in their intrinsic excitability (*Ciarleglio et al., 2015*; *Pratt and Aizenman, 2007*), and behavioral data demonstrate that even young Xenopus tadpoles generate behavioral responses to visual and mechanosensory stimuli (*Dong et al., 2009*; *Roberts et al., 2009*). We used a whole brain ex vivo preparation which allows us to preserve intact the relevant brain circuits (*Wu et al., 1996*). We isolated inputs carrying either visual or mechanosensory information by placing stimulating electrodes in the optic chiasm to activate visual pathways (V) and in the contralateral hindbrain to activate mechanosensory pathways (HB; *Figure 1A*). Previous descriptions of crossmodal integration, evidenced in either the number or temporal distribution of action potentials, reveal that the inter-stimulus interval (ISI) between two crossmodal inputs is a key factor that determines the direction and magnitude of the integrative response (*Meredith et al., 1987*). This temporal relationship can be further modulated by the relative size of the individual responses, where greater multisensory enhancement is observed if the stronger modality precedes the weaker one (*Miller et al., 2015*). To assess how ISI influences the ability of Xenopus tectal neurons to integrate crossmodal stimulus pairs, we performed cell-attached recordings from tectal neuron somata to measure individual spikes evoked by stimulation of sensory inputs (*Figure 1B*). Stimulus intensities were set such that response sizes for each modality were roughly matched. We systematically varied the time interval between the two electrical stimuli delivered to the different sensory pathways. *Figure 1C* shows examples of spike time raster plots at different ISI's in both developmental groups. From these examples two observations become evident. First that paired responses have a faster onset and can have a greater gain than single modality responses. And second that these changes show a temporal dependence on ISI. The temporal dependence of crossmodal responses on ISI is revealed in the population means of the Multi-Sensory Index (MSIn; see Materials and methods) values from both developmental groups (stages 44–46 and stages 48–49; *Figure 1D*). In general, greater and more-positive MSIn

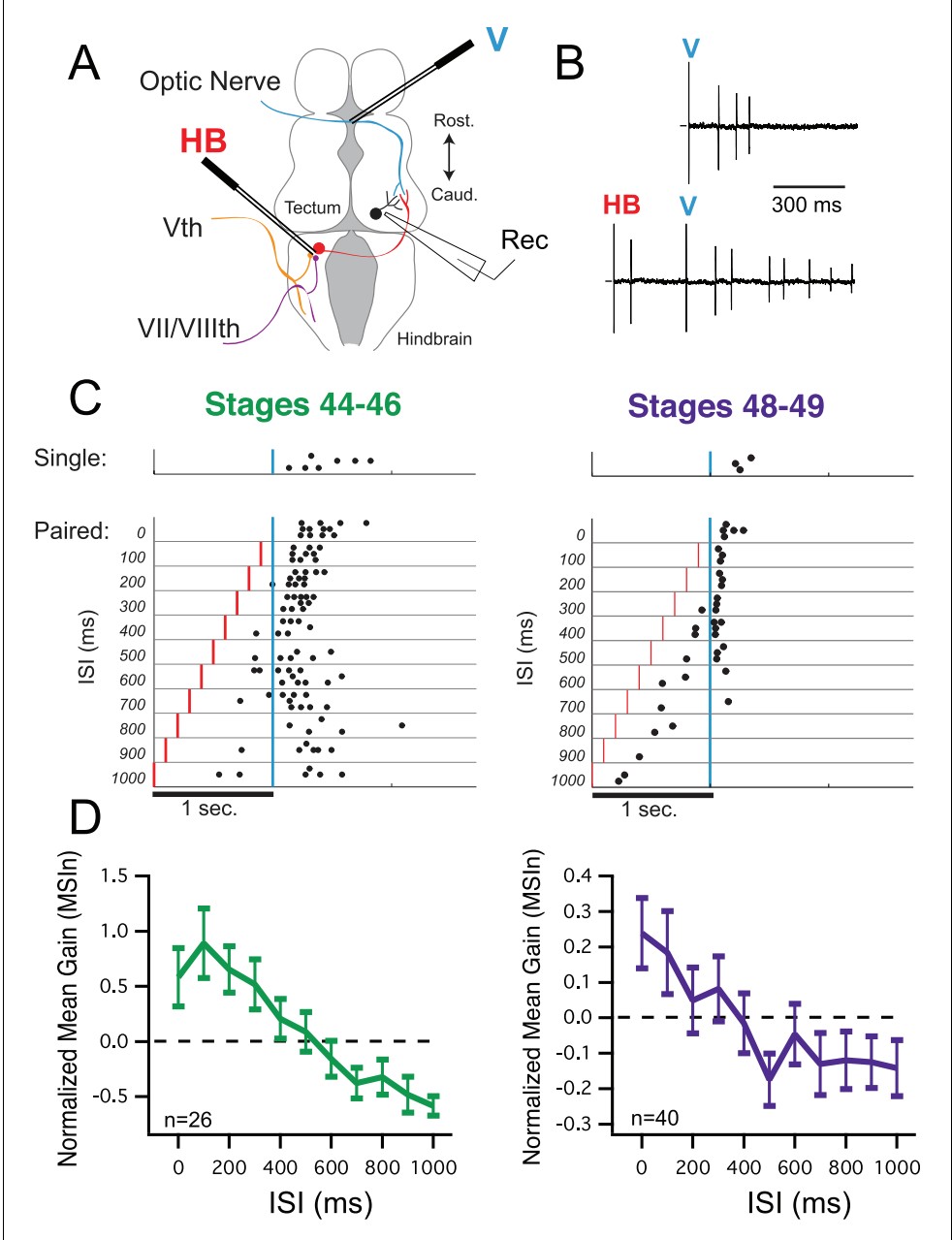

**Figure 1.** Multisensory integration in tectal neurons is dependent on interstimulus interval (ISI). (**A**) Diagram showing placement of the 'V' and 'HB' bipolar electrodes relative to tectal afferents and recording location. (**B**) Raw traces of cell-attached spikes generated during single and paired stimulation. Letters indicate times of stimulus presentation and modality of stimulus. (**C**) Example raster plots from two cells, one from a stages 44–46 animal (*left*) and one from a stages 48–49 animal (*right*) as a function of ISI. (**D**) Grouped data from both developmental groups. In a given cell, data are averaged over trials at each ISI, to determine the MSIn ratio. Plotted here is the population means of these trial-averaged MSIn ratios, at the ISI's tested. Error bars show +/− S. E.M. MSIn = (combined response – single response) / single response.

values were observed as a result of shorter inter-stimulus intervals, whereas negative MSIn values—indicating a suppression of responses due to crossmodal paired stimuli—are produced exclusively by longer inter-stimulus intervals, of up to 1000 ms. Noticeably, the net average MSIn values (both negative and positive) appear to be reduced in the older animals.

One possibility that may explain the apparent decrease in MSIn in older animals is that that at stages 48–49 individual cells could be strongly selective for one or only a few adjacent ISI's, but vary widely with respect to the ISI to which they are most sensitive. As a result, these differences could be drowned out in the population averages. Thus, to investigate the possibility that tectal neurons become tuned to respond best to different crossmodal ISI's over development, we measured the ISI's at which each cell's maximal MSIn values were generated and plotted these ISI values as a histogram for both developmental stages (*Figure 2A,B*). We found that in the stage 48–49 group, maximum MSIn responses occur at a relatively broad range of ISIs (*Figure 2B*) across the population of cells, in contrast to the younger tadpoles, where maximal MSI mostly occurs at short ISIs (*Figure 2A*). Another way to describe this is that in the older tadpoles, tectal neurons, as a population, are tuned to respond optimally to a wider range of crossmodal ISIs (*Figure 2D*; median ISI, st 44–6 = 200 ms, IQR = 300 ms (n=26), st 48–9 = 400 ms, IQR = 775 ms (n=40), p<0.05, Mann-Whitney). However, despite these differences in temporal tuning, the peak MSIn value that could be achieved by a given cell was still roughly twice as large in the younger developmental group (*Figure 2C*; mean max MSI, st 44–6 = 1.15 ± 0.31 (n=26), st 48–9 = 0.55 ± 0.10 (n=40), p<0.05, unpaired T-test).

Next we tested whether any systematic differences between H-V and V-H pairings could be hidden in these combined data. Thus we compared the peak MSIn values at the same ISI, across the two stimulus sequences, and consistently we found that both sequences were equally effective in evoking multisensory responses (see *Table 1*). Likely this is a result of the fact that in these experiments stimulus intensities were adjusted so that response magnitudes were similar across modalities. As a result, we do not distinguish between different stimulus sequences for the remainder of this study.

Independent of the magnitude of a cell's responses to its preferred stimulus, another measure of developmental refinement is the degree of its selectivity for that particular stimulus combination. Because responses to paired stimuli peak at different ISIs in different cells, to study the tuning of ISI selectivity across cells it is necessary to align their MSIn-versus-ISI distributions at the peak MSIn value in each neuron. This alignment of all 'ISI tuning curves' at their maximal MSIn values should provide a common reference point for comparing the shapes of the entire curve, across cells. Tuning curves of individual cells, representing different combinations of H and V offsets, were aligned at their central peak in the X axis to allow for direct comparison of average temporal tuning curve shapes as shown in *Figure 2E*. In the stage 44–6 tadpoles it was evident that across cells, crossmodal response enhancement was present over a wider range of ISIs surrounding the central peak, and that this tuning narrowed significantly over development. To quantify this difference we determined the best curve fit for each data, and the resulting curves were compared statistically using an F test, which tests whether the dataset is best fit by two separate curves or by a single curve. We found significant differences between the tuning curves between developmental stages (F-test: F = 9.966 (2716), p<0.00001), with the curve being narrower in the older group.

Taken together, these data suggest that over development, peak multisensory responses decrease in terms of absolute gain, however cells become preferentially tuned to a greater diversity of temporal intervals, yet also become more sharply tuned around their preferred interval.

## Deviations from linearity

While the MSI metric reveals how the response to a single stimulus is changed by a preceding input, it does not quantitatively address the manner in which the two inputs interact. The method described by Stanford et al. (*Stanford et al., 2005*) allows us to compare the actual output of multisensory responses to what is expected from a purely additive combination of responses to individual, single modality inputs (for details of how the predicted sum is determined, see Materials and methods). Evoked multisensory responses could either result in supralinear, linear or sub linear summation, examples of sublinear and supralinear summation are shown in *Figure 3A*. When we compared the maximal evoked crossmodal response to the response predicted by a linear sum of both inputs, we found that for both developmental stages data points fall below the line of unity indicating a tendency for sub-linear summation (i.e. predicted > actual, *Figure 3B*). This tendency toward sublinearity was evident by plotting a cumulative probability distribution of Z-scores across cells (*Figure 3C*), and by comparing the relative proportion of cells showing sublinear, linear and supralinear responses (*Figure 3D*). Although the cumulative probability curve shifts towards the right

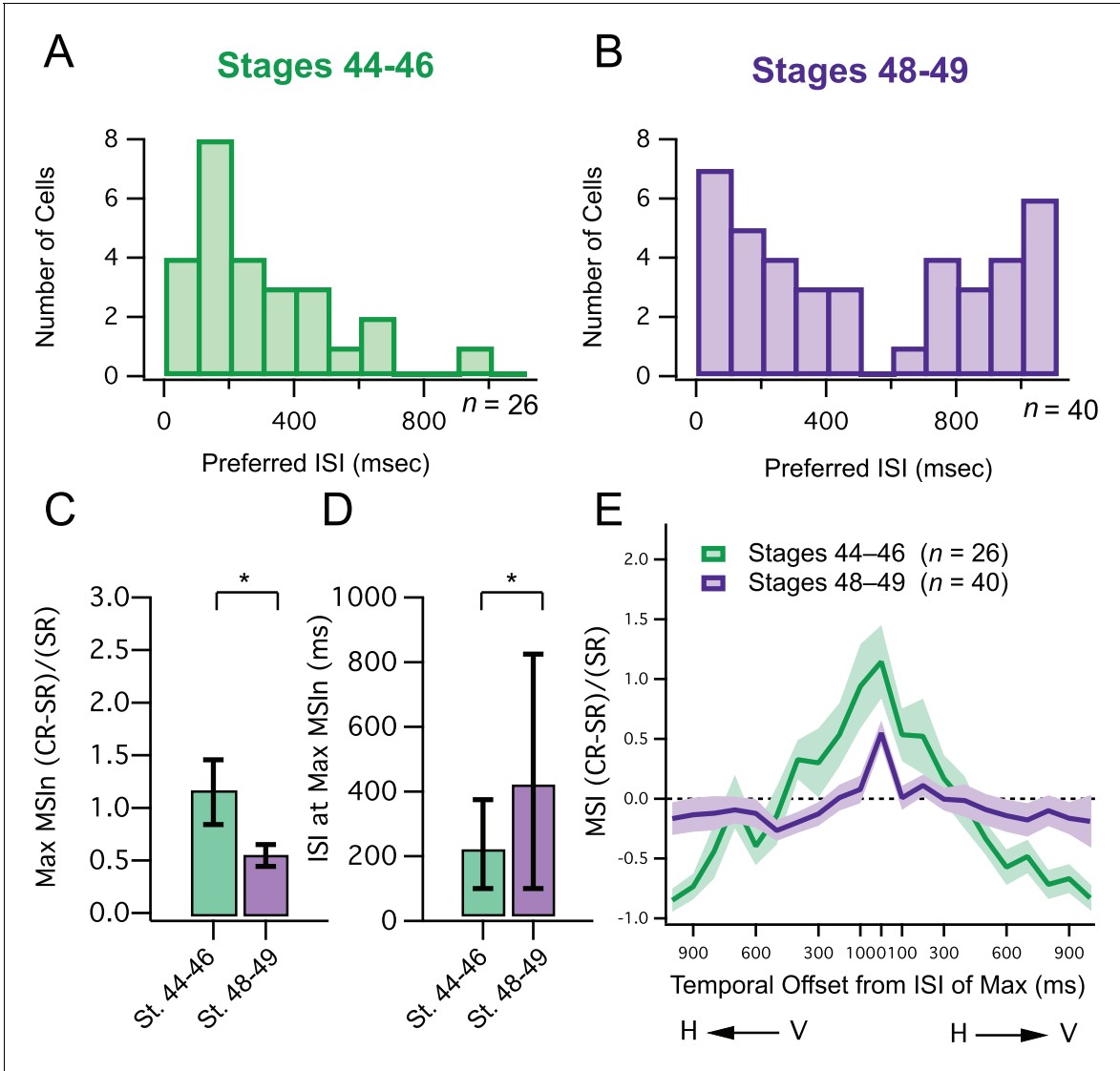

**Figure 2.** Developmental changes in temporal tuning and gain of MSI. (A, B) Histogram bars show the ISI that for each cell, exhibited maximal MSIn values. Notice that values cluster at shorter ISIs in the younger tadpoles. Older tadpoles are tuned to a broader range of ISIs. (C) Maximum MSIn ratios are compared across developmental stages, and show decrease in MSIn gain over development. Error bars indicate +/– S.E.M. (D) The identity of the ISI's responsible for the maximal response (and thus the maximum MSIn ratio) in each developmental group are compared. Error bars show interquartile range. *p<0.05. (E) MSIn-versus-ISI tuning curves from crossmodal pairs were aligned at their peak values, and then averaged across cells. Solid lines connect the population means. Shaded areas demarcate +/– S.E.M. Stage 48–49 cells are more narrowly tuned around their preferred ISI. MSIn = (combined response – single response) / single response.

(supralinear) over development (*Figure 3C*), we found no statistical differences in the distribution of Z scores across developmental groups (Kolmogorov-Smirnov test, D = 0.260, p=0.230), indicating that the overall distributions of integrating responses remains stable during this developmental period. However there was a shift, such that the proportion of cells showing sublinear responses decreased and the proportion showing linear responses increased over development (*Figure 3D*).

## Comparison to unisensory pairs
One possible explanation for the observed developmental decrease in the net gain of multisensory responses and the shortening of the temporal window for multisensory integration could be that these both reflect developmental changes intrinsic to tectal neurons and the tectal network. Between stages 45 and 49, the temporal dynamics in tectal network activity change, such that

**Table 1.** Comparisons between presentation order for cross-modal stimuli do not reveal consistent differences in MSI ratios, in either developmental group. Top: Within both developmental groups, maximum MSIn ratios are compared between the two cross-modal sequences. n.s.: not significant (unpaired t-test). Bottom: The identity of the ISI's responsible for the maximal response is compared between the two cross-modal, within each developmental group. *p<0.05 (Mann-Whitney rank-sum test); n.s.: not significant (Mann-Whitney rank-sum test). V = visual, H = hindbrain

| Crossmodal Order | Stage 44–46 (n=13) | | | Stage 48–49 (n=20) | | |
|---|---|---|---|---|---|---|
| | Mean MSIn ± SEM | p | Summary | Mean MSIn ± SEM | p | Summary |
| VH | 0.98 ± 0.57 | 0.605 | n.s | 0.50 ± 0.17 (n=20) | 0.628 | n.s. |
| HV | 1.31 ± 0.26 | | | 0.60 ± 0.12 | | |
| | Median ISI (ms) ± IQR | | | Median ISI (ms) ± IQR | | |
| VH | 200 ± 250 | 0.964 | n.s. | 350 ± 675 | 0.731 | n.s |
| HV | 100 ± 350 | | | 500 ± 875 | | |

evoked recurrent excitation in the tectum decreases, and becomes more temporally compact (*Pratt et al., 2008*). Furthermore, tectal neurons become less excitable over this developmental time window, generating fewer action potentials when depolarized (*Ciarleglio et al., 2015*; *Pratt and Aizenman, 2007*). If the effects on the temporal properties and raw magnitude of multisensory integration are simply due to changes overall in tectal circuitry and excitability, respectively, then these changes should also be manifested in all paired interactions, including responses to pairs of unisensory stimuli. To test this, we performed paired stimulation of either V or HB inputs over the same range of ISIs we used for the crossmodal pairs. We found that at both developmental stages paired unimodal responses showed significant response enhancement over a wide range of ISIs (*Figure 4A,B*), in contrast to what we had observed for crossmodal responses (*Figure 1D*, *2A*). Moreover, there was significantly more enhancement in response to unimodal pairs than there was for crossmodal pairs, and no developmental decrease was observed in the gain of the paired responses (*Figure 4C*, compare to *2C*; mean max PPE, st 44–6 = 2.25 ± 0.44 (n=29), st 48–9 = 2.57 ± 0.33 (n=26), p = 0.56, unpaired T-test). Similarly, there was also no developmental difference in the range of ISIs that produced the maximal enhancement (*Figure 4D*, compare to 2D; median ISI, st 44–6 = 400 ms, IQR = 300 ms (n=29), st 48–9 = 400 ms, IQR = 300 ms (n=26, p<0.34, Mann-Whitney). Finally, when we looked at the temporal tuning around the maximal ISI, there was no narrowing of the temporal tuning window for unimodal pairs over development as both tuning curves were not significantly different from each other (*Figure 4E*, compare to *2E*, F = 1.394 (2,544), p = 0.249).

Taken together, these data suggest that even at Stage 49, tectal cells can still exhibit robust enhancement to paired stimuli, and thus are not constrained by developmental changes in network dynamics and cellular excitability. In light of this observation, the temporal changes in multisensory integration observed during development likely occur through a different mechanism.

## Inhibition underlies developmental changes in multisensory integration

A second hypothesis for explaining developmental changes in multisensory integration is related to the development of inhibition. In the superior colliculus it has been proposed that maturation of inhibitory feedback correlates with the emergence of multisensory integration. In the tectum, between developmental stages 45 and 49 there is a large increase in evoked inhibitory drive, as well as a negative shift in the Cl- equilibrium potential (*Akerman and Cline, 2006*; *Miraucourt et al., 2012*). Inhibition is known to both narrow the tuning of sensory neurons to their preferred responses and to alter the timing and reliability of sensory driven spike output (*Shen et al., 2011*). Thus, in the tectum, this increased inhibitory drive could constrain multisensory interactions by decreasing the amount of enhancement and narrowing the temporal window for interaction.

If inhibitory inputs are constraining multisensory responses in stage 48–49 tadpoles, then we can make three experimental predictions to test this hypothesis. First, we would predict that blocking inhibition should broaden the temporal tuning for multisensory integration and increase the overall levels of multisensory enhancement. Second, inhibitory blockade should alter the spike timing of

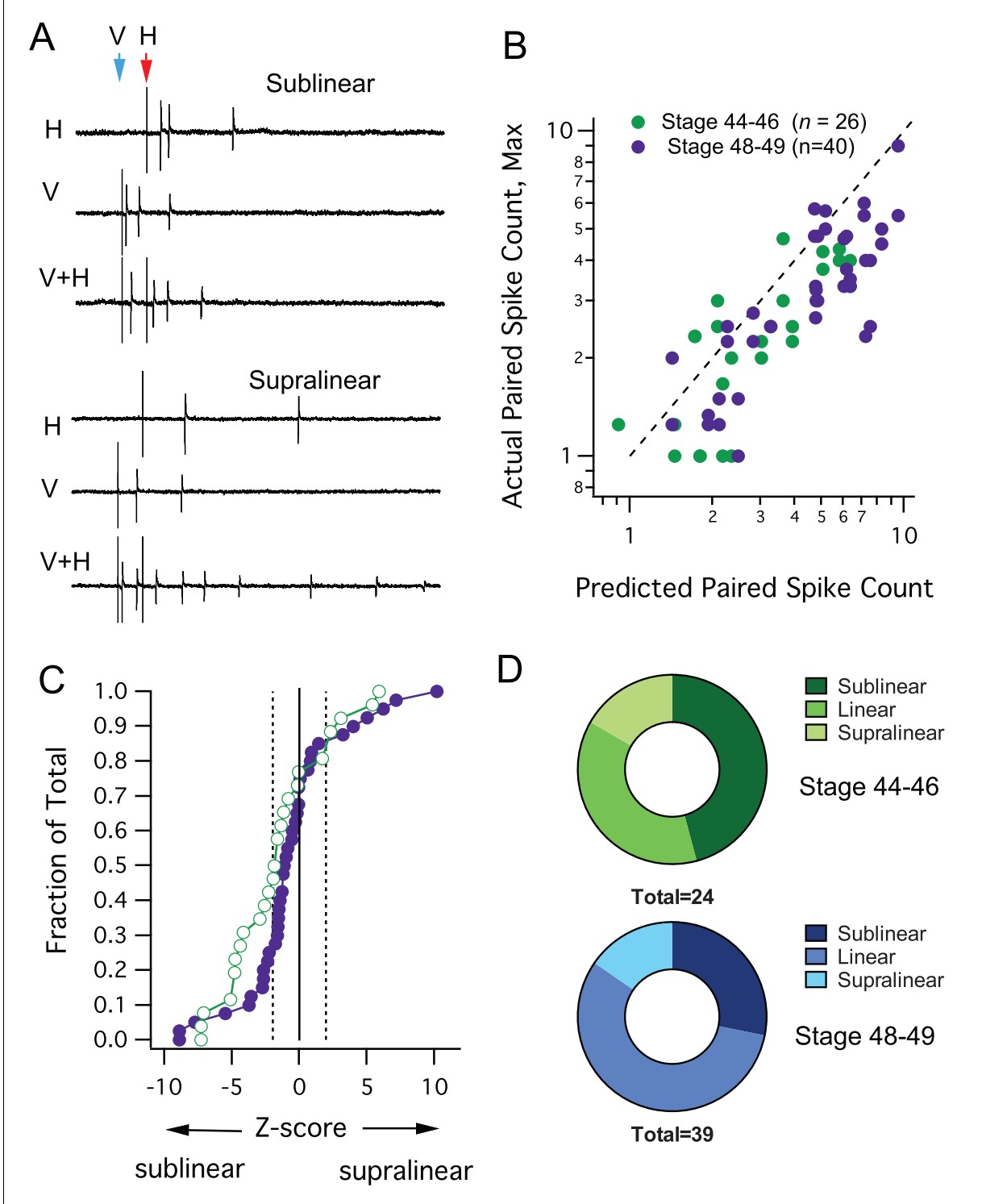

**Figure 3.** Interaction between different modalities is mostly linear or sublinear. (A) Traces show examples of two possible outcomes of interactions between different modality responses. The upper traces show an example of sublinear summation, where the combined response evoked fewer spikes than the sum of the individual responses. The bottom example shows supralinear summation, where the combined response evokes more spikes than the sum of the individual responses. (B) For each cell, the maximum raw spike counts after paired stimulation (combined response) is plotted against the spike count predicted by the sum of adding individual modality responses. Dashed line represents the line of unity. Notice that most points cluster to the right of the dashed line, indicating a sublinear interaction. (C) Cumulative Frequency Distributions of Z-score values are plotted for the comparison between predicted and actual number of action potentials recorded after paired stimulation, in each cell. Vertical dashed lines indicate Z =

*Figure 3 continued on next page*

*Figure 3 continued*

+/–1.97, the point at which actual responses are +/–2 S.D.'s away from the respective predicted response. For greater detail on generation of the predicted responses and calculation of Z-scores, see *Materials and methods*. (D) Distribution of cells by interaction type shows few developmental differences.

multimodal responses, and third, the balance of excitation to inhibition of multisensory responses should decrease over development.

We first tested the prediction that inhibitory blockade in stage 48–49 tadpoles broadens the temporal window for multisensory integration, and increases the overall enhancement in multisensory responses. To block inhibition we performed recordings in the presence of 100 µM picrotoxin, a GABAA receptor blocker. We found that inhibitory blockade resulted in an overall increase in MSIn values over a wide range of ISIs (*Figure 5A*). The ISI values at which the maximum MSIn was recorded were also shorter and became less variable across cells (*Figure 5B,D*; median ISI, PTX = 100 ms, IQR = 425 ms (n=22), no drug = 400 ms, IQR = 775 ms (n=40), p = 0.029, Mann-Whitney) and the overall maximum MSIn per cell increased significantly (*Figure 5C*; mean max MSIn, PTX = 1.96 ± 0.25 (n=22), no drug = 0.55 ± 0.100 (n=40), p = 0.0001, unpaired T-test). Inhibitory blockade also caused a significant broadening of the temporal tuning of the multisensory integration window (*Figure 5E*, F = 28.72(2678), p<0.0001). These effects are all similar to what we had observed in the younger, stage 45 tadpoles without inhibition blocked. Inhibitory blockade also affected the linearity of crossmodal responses. In the presence of picrotoxin, there was an increase in the relative proportion of neurons showing supralinear responses (*Figure 5F*), significantly shifting the cumulative distribution of Z-scores to the right (*Figure 5G*, D = 0.595, p<0.0001, Kolmogorov-Smirnov test). Interestingly, inhibitory blockade did not have any effect on response enhancement caused by unimodal stimulus pairs, suggesting that its effects are not due to general enhancement of network excitability (Max PPR, Control: 2.57 ± 0.32, n=26, PTX: 2.17 ± 0.3, n=12, p=0.39; Difference in Z-scores, D=0.267, p=0.413, Kolmogorov-Smirnov test).

In mammals, spiking responses to crossmodal stimulus pairs always have a more rapid onset, than responses to single stimuli of either modality (*Rowland et al., 2007a*). In our preparation we observe the same effect at both developmental stages and across multiple ISIs. *Figures 6A and B* show the median onset and IQR of spiking responses to both single stimuli and paired stimuli at multiple ISIs for all cells, and we consistently find that paired crossmodal responses are significantly faster than single stimulus responses (*Figure 6D*; st 44–6: p<0.0001, n=25, st 48–49: p=0.04, n= 36, PTX: p<0.0001, n=19, paired T-test). Inhibition in the tectum is known to affect the precision of the spike output pattern of tectal neurons (*Shen et al., 2011*). We find that in picrotoxin, while paired responses still have a faster onset than single responses (*Figure 6C,D*), the onset of paired responses is significantly delayed when compared to stage 48–9 controls (*Figure 6E*; p<0.05). This suggests that inhibition not only regulates the temporal tuning and gain of multisensory integration, but also regulates the timing of tectal cell output.

Finally we tested the third prediction stemming from the hypothesis that enhanced inhibition constrains multisensory responses in older tadpoles: that we would expect to observe differences in the relative amount of inhibition to excitation during cross modal responses at the different developmental time points. We performed whole-cell voltage clamp recordings excitatory and inhibitory synaptic responses during crossmodal stimulation at different ISIs. To isolate excitatory from inhibitory conductances, we recorded at the reversal potential for Cl- (−45 mV) to isolate excitatory currents and at the reversal potential for ionotropic glutamate receptors (+5 mV) to isolate inhibitory currents (*Akerman and Cline, 2006*; *Khakhalin and Aizenman, 2012*). We then converted the synaptic currents into conductance values (see Materials and methods). Sample excitatory and inhibitory conductances are shown in *Figure 7A*, at different ISIs. Overall, we found that inhibitory conductances were much greater in general than excitatory conductances and that the ratio of excitation to inhibition varied greatly across cells, consistent with prior observations (*Akerman and Cline, 2006*; *Khakhalin and Aizenman, 2012*). Based on our previous studies, it is hard to predict a tectal cell's spike output by simply looking at the raw values of individual conductances. Thus it would be more informative to look at the relative enhancement of both excitation and inhibition separately in response to crossmodal pairs. We calculated the MSIn of evoked crossmodal responses to assess

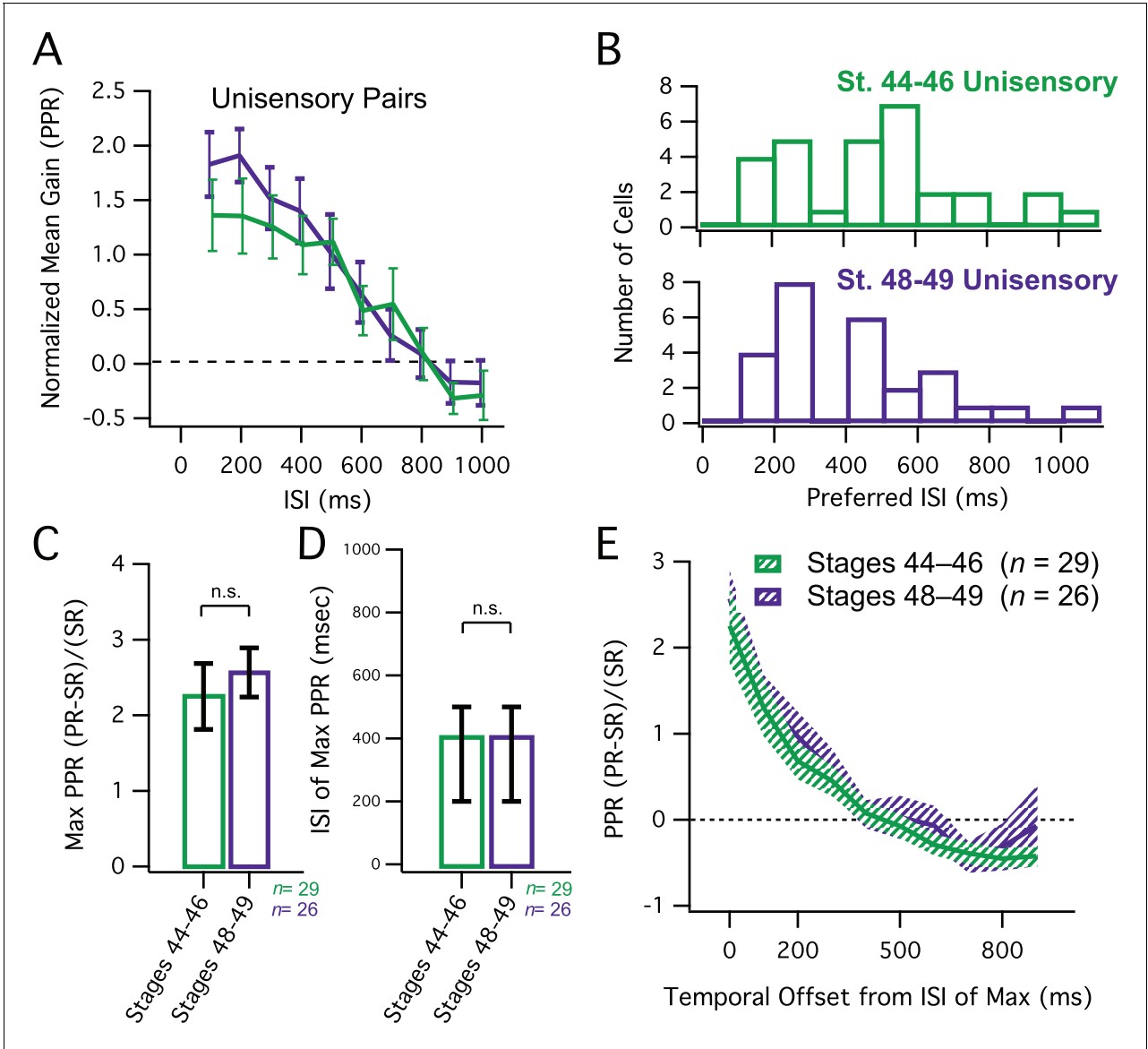

**Figure 4.** Developmental changes in MSI do not simply reflect changes in network dynamics. (**A**) Grouped data showing enhancement of unisensory pairs at both developmental groups. In a given cell, data are averaged over multiple trials at each ISI, to determine the PPR. Plotted here is the population means of these trial-averaged PPR ratios, at the ISI's tested. Error bars show +/– S.E.M. (**B**) Histogram bars show the ISI that for each cell, exhibited maximal PPR values. Notice that values tend to cluster at shorter ISIs in both groups. (**C**) Maximum PPR ratios are compared across developmental stages, and show no change in PPR gain over development. Error bars indicate +/– S.E.M. (**D**) The identity of the ISI's at which maximal PPR occurred, separated by developmental group. Error bars show interquartile range. n. s. = not significant. (**E**) PPR-versus-ISI tuning curves from unimodal pairs were aligned at their peak values, and then averaged across cells. Since pairs were unisensory, only one direction is shown. Solid lines connect the population means. Shaded areas demarcate +/– S.E.M. Tuning for unisensory pairs does not change over development. PPR = (paired response – single response) / single response.

the amount of response enhancement of both excitation and inhibition at the different ISIs (see Materials and methods). At both developmental stages we found on average an overall enhancement of both excitatory and inhibitory multisensory responses across all ISIs (*Figure 7B,C*). However we found proportionally greater enhancement across ISIs of excitatory responses in the stage 44–46 tadpoles, compared to stage 48–49 ($p=0.037$, 2-way ANOVA). In contrast we found that enhancement of inhibitory conductances, and its dependence on ISI, did not change over this developmental time window. We also compared data from individual cells by plotting the maximal Excitatory MSIn

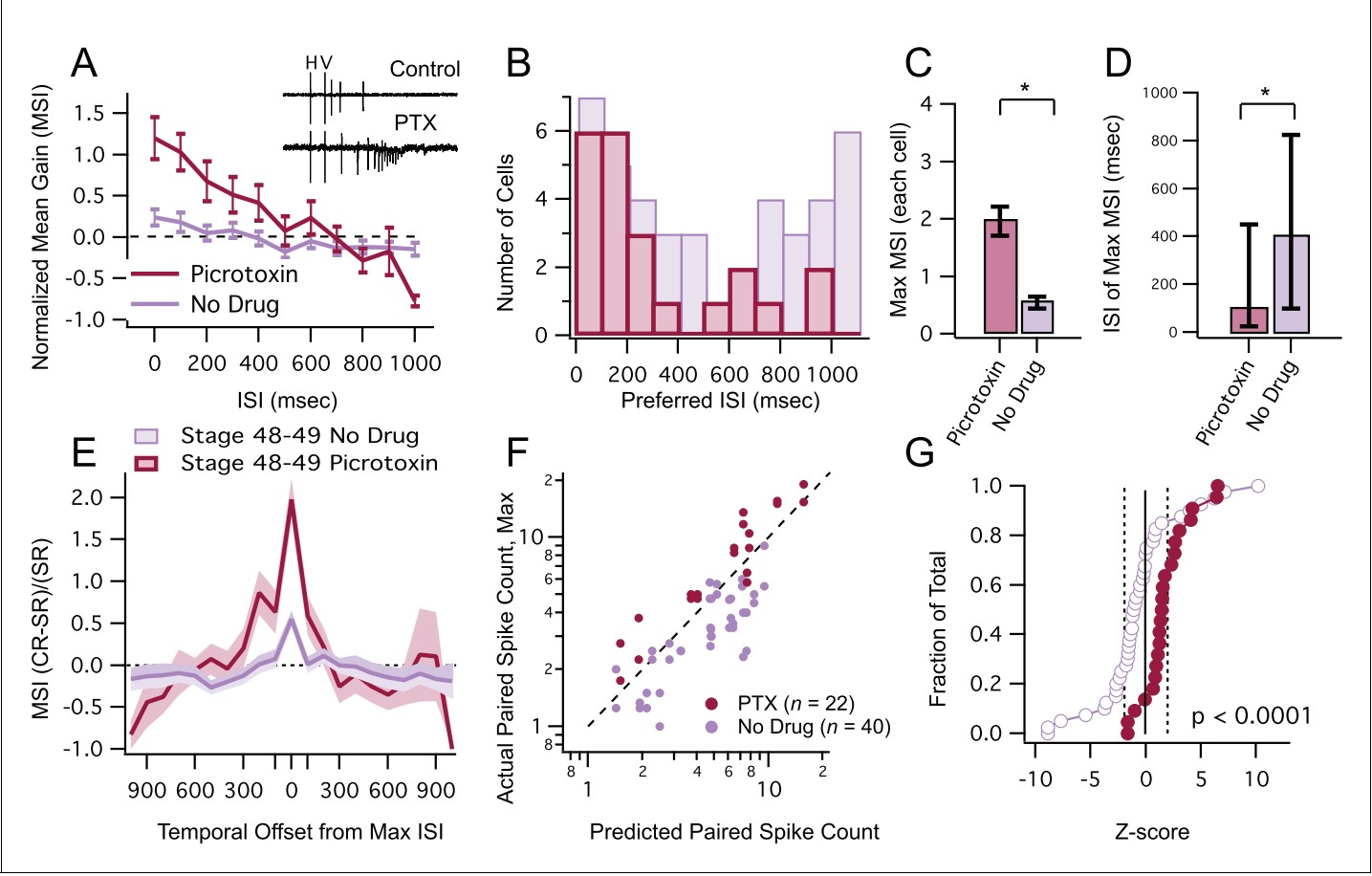

**Figure 5.** Inhibitory blockade broadens tuning window and enhances gain of MSI. (**A**) Grouped data from stage 48–9 tadpoles in control condition and in the presence of GABAA receptor blocker PTX. In a given cell, data are averaged over trials at each ISI, to determine the MSI ratio. Plotted here are the population means of these trial-averaged MSIn ratios, at each ISI tested. Error bars show +/– S.E.M. (**B**) Histogram bars show the ISI at which each cell exhibited maximal MSIn values in control and PTX groups. Notice that PTX shifts the preferred ISI values to shorter intervals. (**C**) Maximum MSIn ratios are compared across control and PTX conditions, and show increase in MSI gain with inhibitory blockade. Error bars indicate +/– S.E.M. (**D**) The identity of the ISI's responsible for the maximal response (and thus the maximum MSI ratio) in control and PTX groups are compared. Error bars show interquartile range. *p<0.05. (**E**) MSI-versus-ISI tuning curves from crossmodal pairs were aligned at their peak values, and then averaged across cells. Solid lines connect the population means. Shaded areas demarcate +/– S.E.M. PTX treated cells are more broadly tuned around their preferred ISI. (**F**) For each cell, the maximum raw spike counts after paired stimulation (combined response) is plotted against the spike count predicted by the linear sum of individual modality responses. Dashed line represents the line of unity. Notice that in PTX treated group most points cluster to the left of the dashed line. (**C**) Cumulative Frequency Distributions of Z-score values are plotted for the comparison between predicted and actual number of action potentials recorded after paired stimulation, in each cell. Vertical dashed lines indicate Z = +/–1.97, the point at which actual responses are +/–2 S.D.'s away from the respective predicted response. For greater detail on generation of the predicted responses and calculation of Z-scores, see *Materials and methods*. Z-scores in the PTX treated group are significantly shifted toward the right (p<0.0001, Kolmogorov-Smirnov test).

vs. the corresponding Inhibitory MSIn for both developmental stages (*Figure 7D*). In both cases maximal Excitatory and Inhibitory MSIn values were positively correlated (St 44–46: r = 0.758, p<0.0001; St 48–49: r=0.767, p<0.0001). However the ratio between excitatory enhancement and inhibitory enhancement was greater in younger tadpoles than in older tadpoles, overall, consistent with the observation that excitatory currents show proportionally greater MSIn values than inhibitory currents (test for differences in slopes of linear fits: p=0.023, F = 5.56(1,45).

Taken together, these data indicate that in younger tadpoles, the ratio of excitation to inhibition, in response to crossmodal stimulus, is greater than in the older group, which may in part explain the greater amount of multisensory enhancement of spike output observed in the younger tadpoles. This change in the excitation to inhibition ratio may also explain the sharpening of the temporal tuning of multisensory responses. However, these data do not directly account for the dependence of

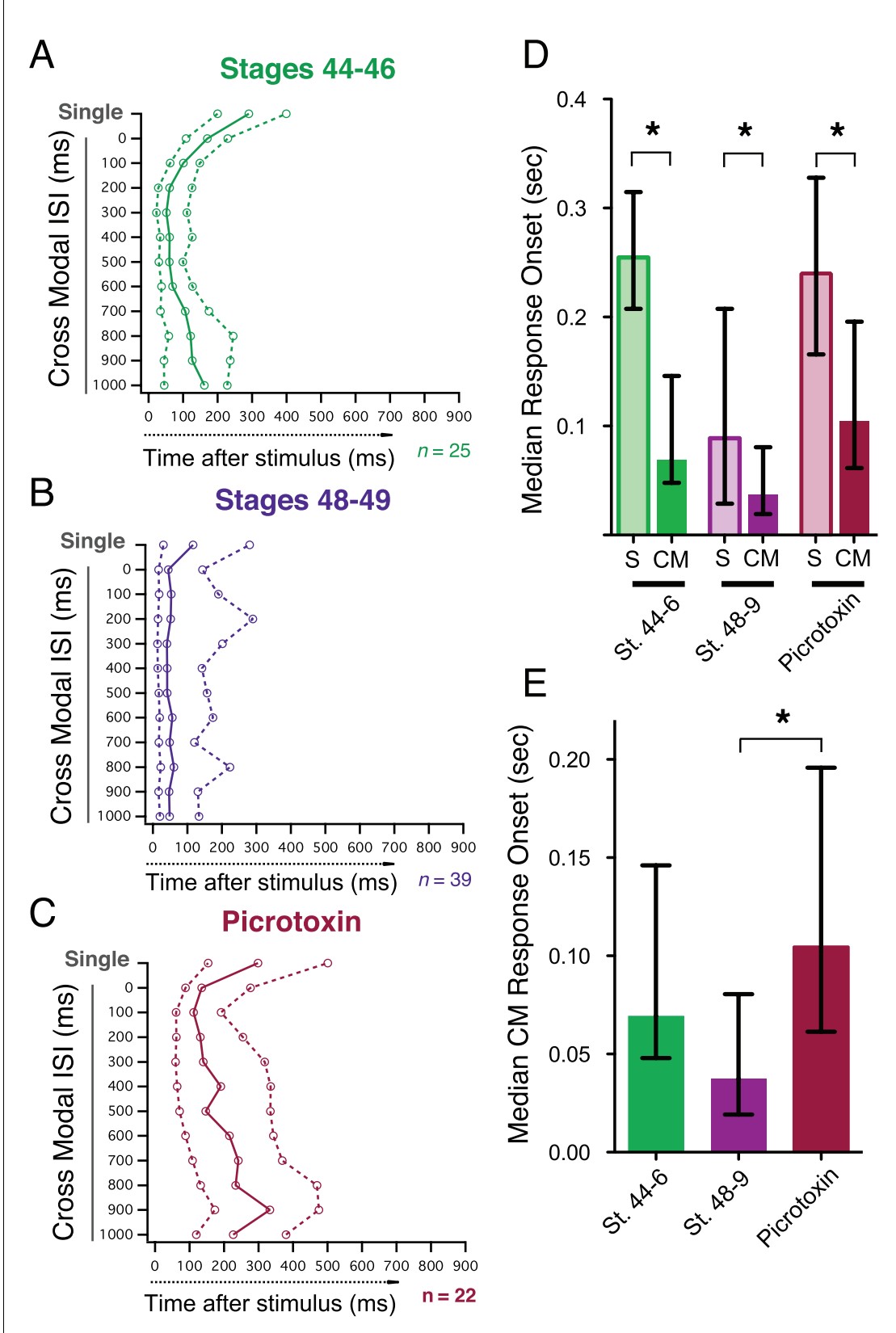

**Figure 6.** Crossmodal responses have faster onset latencies than single unimodal responses across groups, but inhibitory blockade slows response onset times. (A, B, C) Plot of response onset times for single responses and crossmodal responses at multiple ISIs for all experimental groups. For each
*Figure 6 continued on next page*

*Figure 6 continued*

paired ISI, as well as the appropriate control stimulus (*left axis*), the. 25,. 50 (median), and. 75 quartiles of post-stimulus times for all first spikes are plotted on the bottom axis. Dashed lines connect 0.25 and 0.75 quartile values, and solid lines connect. 50 quartile (median) values. (D) Comparison of response onset times of paired vs. unpaired responses shows that across conditions, cross modal responses occur faster than single modality responses. (E) Comparison between paired response onset times across groups shows that while there is no significant speeding up of the response over development, picrotoxin delays onset times. Bars indicate median values, error bars show the IQR. *p< 0.05

multisensory integration on ISI. One possibility is that the temporal dynamics of concurrent excitation and inhibition may vary as a function of ISI, thus resulting in different patterns of spike output. In order to test this hypothesis we would need to record excitation and inhibition simultaneously during a single response, due to the trial-to-trial variability in response shape, something that was not done in the present study.

## Discussion

In this study we provide novel insight into the synaptic and circuit-level mechanisms that underlie developmental changes in the temporal rules mediating multisensory integration in the optic tectum. We found that optic tectal neurons exhibit enhanced responses to crossmodal stimulus pairs, compared to single modality responses, and that this enhancement is heavily dependent on the ISI. The temporal window for multisensory enhancement becomes narrower between developmental stages 44–46 and 48–49, but across the tectal population neurons become preferentially tuned to a wider range of ISIs. We also find that the net gain of multisensory responses decreases between these stages. Across both developmental stages we also consistently observed a speeding up of response latency in paired compared to single responses. Inhibitory blockade in older tadpoles results in a broadening of the integration window, increased gain, and delays in response latency. Furthermore, the balance of excitation to inhibition during crossmodal responses decreases over this developmental period.

Taken together our data are consistent with a model in which crossmodal inputs to the tectum recruit local networks of excitatory neurons, resulting in enhanced tectal cell responses. Crossmodal inputs also recruit local inhibition, and the relative enhancement of inhibition, versus excitation, increases over development. This change in relative enhancement sculpts the temporal sensitivity of the tectal network such that individual neurons become more narrowly tuned to specific ISIs, thus enhancing the diversity of responses to different temporal combinations of inputs. Local inhibition also serves to constrain the timing of evoked multisensory responses, by limiting the temporal window in which spikes can occur during a response, and thus shortening response latencies.

Our data are consistent with this model in several ways. First we show that pharmacologically blocking inhibition results in a broadening of the window for multisensory integration in the older tadpoles. During inhibitory blockade, tectal cells are no longer narrowly tuned to a specific ISIs, respond with a greater gain, and preferentially respond to shorter ISIs. All of these characteristics are similar to what we observed in the younger tadpoles. Second, we find that the relative amount of multisensory enhancement (as measured by higher MSIn values) of inhibitory to excitatory responses is greater in the older tadpoles, indicating that they receive more inhibition relative to excitation. There is ample evidence throughout the nervous system that suggests that the inhibition to excitation ratio can sharpen the tuning of sensory neurons to their preferred stimulus (*Pouille and Scanziani, 2001*; *Priebe and Ferster, 2008*; *Shen et al., 2011*; *Wehr and Zador, 2003*). Having more inhibition can limit the range of stimuli that can effectively drive a cell above spike threshold. Our observation that the balance of excitation and inhibition changes developmentally for multisensory responses is consistent with this view.

Inhibitory blockade also results in alterations of how crossmodal inputs summate, by causing an increase in the proportion of neurons that show supralinear summation. Our data show that at both developmental stages tectal neurons tend to either mostly show linear or sublinear summation, with a small percentage of cells showing supralinear responses. Thus it is likely that inhibition is also constraining, to a degree, the amount of multisensory integration in the younger tadpoles. Finally, blocking inhibition delays the median onset latency of multisensory responses, which is consistent

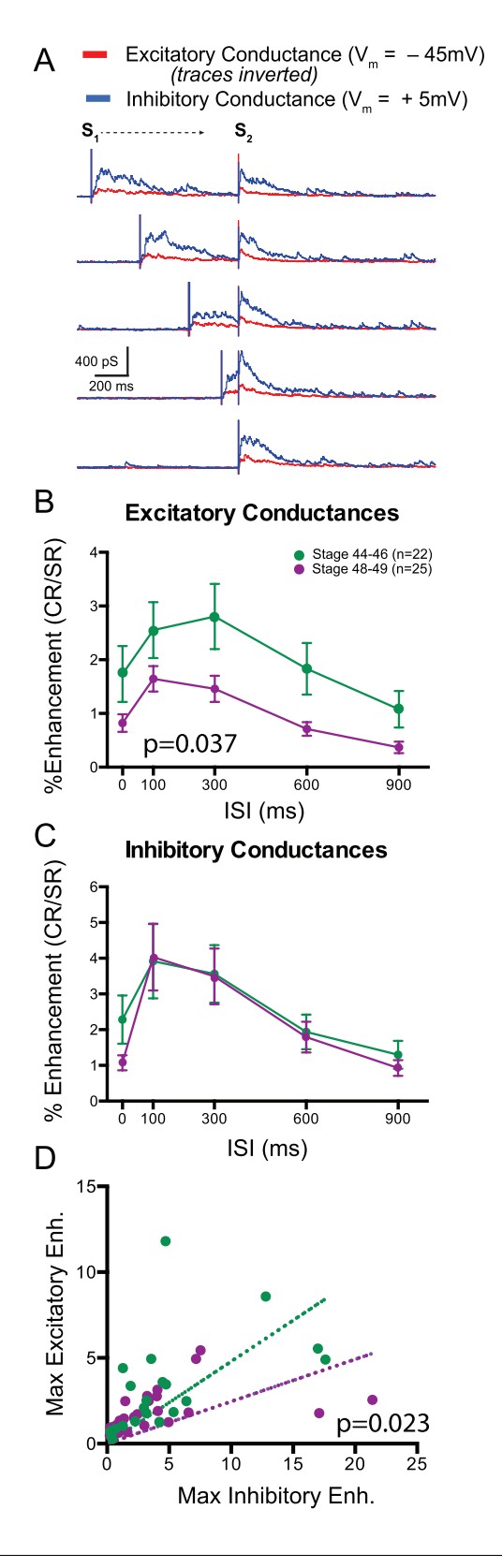

**Figure 7.** Multisensory responses have a greater inhibitory to excitatory ratio in older animals. (**A**) *Figure 7 continued on next page*

with a role for feedforward and feedback inhibition in controlling the temporal output of neuronal responses of sensory neurons (*Pouille and Scanziani, 2001*; *Shen et al., 2011*). Although inhibitory feedback has been proposed to be a central mechanism for regulating multisensory integration in the superior colliculus (*Rowland et al., 2007b*), to our knowledge ours is the first experimental evidence directly supporting this model.

Recent work by *Miller et al. (2015)* shows that the temporal tuning of crossmodal responses can be strongly dependent on input size disparity, such that if the stronger modality is presented first, there will be a greater amount of multisensory enhancement than if the weaker modality is presented first. When both modalities are matched in their response sizes, then the temporal order becomes less important. This 'stronger-first' rule is thought to result from differential recruitment of inhibition. In our findings we did not observe any differences between the order in which visual or hindbrain inputs were activated, likely because we used responses of similar size in our experiments. It will be interesting to see in future studies if the 'stronger first' rule is preserved in the tectum, and whether this rule can be disrupted by inhibitory blockade.

Our findings in the optic tectum also share other fundamental similarities and differences with other animal models of multisensory integration. (i) In the superior colliculus of mammals (*Meredith et al., 1987*) as well as in the optic tectum of owls (*Bergan and Knudsen, 2009*), there are clear demonstrations of dependency of multisensory integration to the temporal interval between different modality inputs. Furthermore, a developmental narrowing of the temporal window underlying multisensory integration has been shown behaviorally in humans (*Hillock et al., 2011*) and has been proposed to mediate development of important cognitive processes such as temporal binding. (ii) Second, like in the tadpole tectum, in mammals, collicular cells show also differences in the type of integration, having a combination of supralinear, linear and sublinear responses (*Alvarado et al., 2007*). (iii) Third, another important aspect that underlies the development of multisensory integration in the mammalian colliculus is the presence of cortical feedback (*Jiang et al., 2001*; *Wallace et al., 1993*; *Wallace and Stein, 2000*). While tadpoles do not have a proper cortex, during the developmental

*Figure 7 continued*

Sample excitatory (red) and inhibitory (blue) synaptic conductances recorded at different ISIs. S1 and S2 show timing of the individual modality responses (V and HB, respectively in this example). (B) Plot of crossmodal enhancement of excitatory synaptic conductances as a function of ISI at both developmental stages. Notice that excitation becomes relatively less enhanced in the older tadpoles. (C) Plot of crossmodal enhancement of inhibitory synaptic conductances as a function of ISI at both developmental stages. Notice that there is no developmental change in the amount of inhibitory enhancement. P value determined from 2-way ANOVA. (D) Plot of maximum excitatory and inhibitory enhancement for each cell. Dotted line represents linear fit for each developmental group. p value determined by an F-Test and compares both fits. Notice that older cells exhibit relatively more inhibitory than excitatory enhancement.

stages used in this study tadpoles do have forebrain feedback into the tectum (*Zittlau et al., 1988*), as well a local recurrent circuits (*Pratt et al., 2008*) which can provide the necessary excitatory network activity to drive multisensory integration in individual neurons. The relative contributions of both of these feedback circuits on multisensory integration remains to be determined, yet based on the timing of the tectal neuron spiking, it is likely that it is this recurrent activity that drives the majority of spiking activity, rather than the afferent input itself (*Pratt et al., 2008*). Since the temporal output characteristics of these recurrent circuits has been shown to be highly plastic in response to experience (*Pratt et al., 2008*), this raises the possibility that this network plasticity can help tune the selectivity of different tectal cells over development to specific combinations of crossmodal ISIs. This type of network plasticity may also help explain observations in the mammalian superior colliculus where temporal statistics of cross-modal stimuli during development and in adulthood shape the temporal tuning of multisensory responses (*Xu et al., 2012*: *Yu et al., 2013*). Using the Xenopus tadpole tectum preparation it should be possible to study this type of plasticity at the cellular level.

Our findings also show some important differences with the mammalian literature. (i) In the present study we find that pairs of unimodal stimuli result in greater integration over a wider range of intervals than crossmodal pairs. This finding is opposite to what is observed in mammals, where crossmodal inputs are more effective than unimodal ones (*Alvarado et al., 2007*). There are likely two mechanisms underlying this difference. The first is due to presynaptic interactions that lead to paired pulse facilitation, which has been described for both visual and hindbrain inputs in the tadpole tectum (*Deeg et al., 2009*). The second may be due to localization of inputs in the dendritic arbor of tectal neurons. Unisensory pairs thus, might be more effective in driving local dendritic depolarization since they are not spatially segregated, and therefore activate non linear response elements such as NMDA receptors or voltage gate Ca++ channels, thus enhancing the amount of integration. (ii) A second key difference is that the developmental decrease in pair-driven enhancement of excitatory conductances and spike output appears to contradict developmental studies in mammals, where the progressive refinement of superior colliculus receptive fields is associated with the capacity for response enhancement, after paired stimulation (*Wallace and Stein, 1997*). Indeed, an analogous developmental refinement of visually-driven receptive fields, specifically, is found for both excitatory and inhibitory conductances—recorded in the same neuron—in the X. laevis optic tectum over the same developmental stages examined here (*Tao and Poo, 2005*). One possible explanation for this discrepancy is the use of bipolar stimulating electrodes in our preparation to evoke action potentials throughout an afferent pathway. While this stimulation paradigm provides excellent temporal control of stimulus delivery, this stimulation method by its nature has no selectivity for spatiotopic organization. Each stimulus is therefore considered 'whole-field', and loses its spatial specificity. Furthermore, in vivo stimuli are likely to also have more temporally complex patterns that may result in somewhat different multisensory interactions. It will be important in future studies to compare the rules underlying multisensory integration using in vivo visual and mechanosensory stimuli.

A second possible explanation to this discrepancy between tadpole tectum and mammalian colliculus, is that in tadpoles tectal neurons are innervated by sensory input from very early developmental stages (*Gaze et al., 1974*), and thus the developmental stages of the circuits in this study may correspond to a much younger stage than what is typically looked at in mammalian studies.

Overall, our findings use a reduced preparation to describe, for the first time at the cellular level, the development the temporal properties underlying multisensory integration in the Xenopus

tadpole optic tectum. We have used this preparation to uncover some of the basic phenomenology of multisensory integration in this preparation, and have shown that recruitment of recurrent inhibition is a central mechanism in mediating this process. We have also shown that some of the basic features and computations of multisensory integration observed in higher order mammalian preparations are conserved in the developing amphibian tectum, indicating that these are likely to be fundamental and evolutionarily conserved processes. These studies further open the door to follow up studies focusing on the behavioral consequences of these processes, as well as on the role of early sensory experience in determining the temporal properties of multisensory integration. Several neurodevelopmental disorders including autism, are often accompanied with deficits in sensory integration (*Baum et al., 2015*; *Markram and Markram, 2010*; *Foxe et al., 2015*). Thus, these results will not only enhance our general understanding of the development and merging of neural circuits and systems, but they will better allow us to understand how these processes may go awry in neurodevelopmental disorders.

## Materials and methods

### Experimental animals

Wild-type Xenopus laevis tadpoles were raised on a 12 hr light/dark cycle at 18°C in 10% Steinberg's solution. In our laboratory, tadpoles reach the Nieuwkoop and Faber (*Nieuwkoop and Faber, 1956*) developmental stages 44–46 at 7–10 dpf, and stage 48–49 at 14 dpf. Xenopus laevis tadpoles between developmental stages 44 and 49 were utilized for this study. The Brown University Institutional Animal Care and Use Committee (IACUC) approved all handling of animals in accordance with National Institutes of Health (NIH) guidelines.

### Whole-brain preparation

The whole-brain preparation is as follows, after *Wu et al. (1996)*: animals were first anesthetized in 0.01% tricaine methane sulfonate (MS-222) in 10% Steinberg's, the dorsal surface of skin was then removed to expose the brain, the dorsal midline was cut at all levels from base of spinal cord through the olfactory bulbs, and the brain dissected out. The preparation was transferred to a recording chamber with room temperature HEPES-buffered extracellular saline (containing: 115 mM NaCl, 4 mM KCl, 3 mM $CaCl_2$, 3 mM $MgCl_2$, 5 mM HEPES, and 10 mM glucose; pH 7.2, 255 mOsm) and positioned on top of a block of Sylgard, with the exposed walls of the ventricle facing upwards. Shortened insect pins were inserted through the caudal extent of hindbrain and through one or both olfactory bulbs. For stimulation of retinal ganglion cell axons, a bipolar stimulating electrode consisting of two adjacent 25-μm platinum leads (CE2C75; FHC, Bowdoin, ME) was placed at the optic chiasm (OC), and for stimulation of mechanosensory projections, a second bipolar stimulating electrode was placed in the rostral hindbrain (HB) contralateral to the recording site (see *Figure 1*). Individual neurons in the optic tectum were visualized through a light microscope with a 60× water-immersion objective, in combination with an infrared CCD camera. To achieve access to the tectal cells at the recording site the jagged tip of a broken glass micropipette was used to lift away the periventricular membrane, with the aid of a micromanipulator. These recording sites were selected consistently from within in the middle third of the optic tectum's rostral-caudal dimension, to avoid introducing variability in the maturational state of neurons studied at a given stage of tadpole development, given that the tectal circuit matures along a rostral-to-caudal gradient in individual animals (*Pratt et al., 2008*; *Wu et al., 1996*).

### Electrophysiology

Glass micropipettes were pulled for tip resistances of 8–12 MΩ. For whole-cell recording micropipettes were filled with filtered $Cs^+$-methane sulfonate/TEA intracellular saline (containing: 80 mM $Cs^+$-methane sulfonate, 20 mM TEA, 5 mM $MgCl_2$, 20 mM HEPES, 10 mM EGTA, 2 mM ATP, and 0.3 mM GTP; pH 7.2, 255 mOsm), and for loose cell-attached (LCA) recording of action potentials, the same micropipettes were filled with filtered extracellular saline. In experiments where blocking $GABA_A$-receptor mediated inhibition was required, 0.1 mM picrotoxin was added to the extracellular solution. Electrophysiological signals were detected with an Axopatch 200B amplifier, digitized at 10 kHz by a Digidata 1322A analog-to-digital converter, and formatted for recording by pClamp 9

acquisition software. Leak subtraction was performed on-line, in real-time by the acquisition software. In our recording conditions the junction potential is predicted to be 12 mV, but was uncorrected in the recorded traces. To detect changes in access resistance over the course of a recording, a 5 mV depolarizing square wave was applied at the start of each trace. In all experiments, only cells demonstrating responses to both optic chiasm and hindbrain stimulation were chosen for recording.

Loose-cell attached recordings were used to measure action potentials without breaking through the cell membrane and without electrical access, and were defined as having seal resistances in the 40–200 MΩ range. The pipette tip was dirtied prior to cell contact to prevent formation of a tight seal. Action potentials were detected off-line by importing the digitized traces into the AxoGraphX analysis environment and by using an amplitude threshold to identify events and determine post-stimulus onset times.

Experiments to examine the temporal characteristics of excitatory and inhibitory synaptic conductances were performed with whole cell recordings. Voltage-clamp mode was used to isolate synaptic conductances mediated by excitatory neurotransmitter receptors and those mediated by inhibitory neurotransmitter receptors. By using the voltage-clamp to hold the cell's membrane potential at the reversal potential of a given synaptic current, it is possible to eliminate the driving force on the ions mediating that current and thus 'zero' the amplitude of that particular type of synaptic event. Previous work in the laboratory has shown that the reversal potential of excitatory AMPA and NMDA receptor-mediated currents is +5 mV and the reversal potential of synaptic currents mediated by inhibitory GABA$_A$ receptors is –45 mV (*Bell et al., 2011*).

## Stimulus properties

Electrical stimulation was initiated automatically by the acquisition software. At pre-specified time points, ISO-Flex stimulus isolators (AMPI, Jerusalem, Israel) were activated for 0.2 ms by an ON-OFF command signal from the digitizer. The output of each stimulus isolator (one for each stimulus electrode) was manually set, based on the responsiveness and dynamic range of each cell, to deliver between 10 μA and 800 μA across the poles of it's bipolar electrode, for the duration of the command signal. Stimulus strength was set to ensure that spike output from both inputs was closely matched.

## Data analysis

All analyses were performed offline, using AxoGraphX software and the MATLAB programming environment. Prism software (GraphPad) was used for curve fitting and statistical tests. Sample sizes were based on power analyses and known variability from prior work in our experimental system. Statistical tests used, p and N values are indicated within the results. Non-parametric tests were used when appropriate. No outliers were removed from the data, and all multiple comparisons were corrected using a Bonferroni correction. Determination of the Multisensory Index was performed as described by Meredith and Stein (*Meredith and Stein, 1983*), using the equation: MSIn = (CM – SM)/SM. where SM is the average number of action potentials evoked by a single-stimulus presentation (<u>S</u>ingle <u>M</u>odality), CM is the average number of action potentials evoked by the paired-stimulus presentation (<u>C</u>ombined <u>M</u>odality), and MSIn is the <u>M</u>ultisensory <u>I</u>ndex. This measure indicates the difference (or 'gain') in a cell's response (positive or negative) resulting from the addition of a second stimulus, corrected for the cell's maximal single-stimulus response. Because MSIn corrects for each neuron's maximal output in the single-stimulus condition, it is a normalized measure that can be compared across neurons that may vary in overall spike output. In this study MSIs were calculated separately for both sensory modalities and combinations of temporal sequences. Determination of the predicted neuronal response to paired stimuli, both unimodal and crossmodal, was performed in the manner of Stanford and colleagues (*Stanford et al., 2005*). For each cell, responses after each trial of the individual (baseline) stimulus presentations were collected and, as appropriate for the type of paired responses being predicted, all possible unimodal trial-by-trial combinations or all possible crossmodal trial-by-trial combinations were determined. For each such combination of trials, the sum of spike counts recorded in each was calculated. Thus, with 33 single-stimulus trials delivered through each modality, in each cell, 33 × 33 = 1089 possible sums exist. In the actual experiments, however, at each ISI 4 trials of paired stimuli were presented, and the mean response over these trials was determined. To mirror these procedures in the predictive analysis, for each cell and

for each type of stimulus pair, 4 of the possible sums were randomly selected (with replacement) and their mean taken. This randomly sampling and averaging was repeated 10,000 times for each pair type, to create an approximately normal distribution of predicted mean sums. In each cell, Z-score comparisons of the actual mean response, at each ISI of each pair type, were then performed against this distribution.

## Acknowledgements

The authors would like to thank Irina Sears and Phouangmaly Mimi Oupravanh for animal and lab care. This work was supported by NSF IOS-1353044 (CDA), NEI 5T32-EY018080 (DLF, PI: MJ Paradiso) and a Fox Postdoctoral Fellowship from Brown University (ASK).

## Additional information

### Funding

| Funder | Grant reference number | Author |
|---|---|---|
| National Science Foundation | NSF IOS-1353044 | Daniel L Felch<br>Arseny S Khakhalin<br>Carlos D Aizenman |
| NIH Office of the Director | NEI 5T32-EY018080 | Daniel L Felch |
| Brown University | Fox Postdoctoral Fellowship | Arseny S Khakhalin |

The funders had no role in study design, data collection and interpretation, or the decision to submit the work for publication.

### Author contributions

DLF, Conception and design, Acquisition of data, Analysis and interpretation of data, Drafting or revising the article; ASK, CDA, Conception and design, Analysis and interpretation of data, Drafting or revising the article

### Author ORCIDs

Arseny S Khakhalin, http://orcid.org/0000-0002-0429-1728
Carlos D Aizenman, http://orcid.org/0000-0002-7378-7217

### Ethics

Animal experimentation: The Brown University Institutional Animal Care and Use Committee (IACUC) approved all handling of animals in accordance with National Institutes of Health (NIH) guidelines. Experiments were performed under IACUC protocol #1308000008C002, most recently renewed August 12, 2015.

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
