## [Decision Letter]

Thank you for submitting your article "Multisensory Integration in the Developing Tectum is Constrained by the Balance of Excitation and Inhibition" for consideration by *eLife*. Your article has been reviewed by two peer reviewers, including Hollis Cline, and the evaluation has been overseen by Gary Westbrook as the Senior Editor. The reviewers have discussed the reviews with one another and the Senior Editor has drafted this decision to help you prepare a revised submission.

Summary:

The reviewers nicely summarize the strengths of the manuscript as below. However, the authors should address the comments (with discussion or data) under "essential revisions" in preparing the revised manuscript.

Essential revisions:

1) In the first paragraph of the Results the authors refer to previous studies of ISI in mammalian systems. The scholarship here is evolving and there is a more nuanced view of these effects and the consequences of stimulus order on the resultant response (e.g., see Miller et al., J. Neuroscience 2015).

2) In the fifth paragraph of the Discussion they make a point about the sensitivity of timing to experience in this circuit. This is an excellent point about how the circuit involved adapts to its use. This system is ideal for exploring this issue in more detail. To my knowledge only two such studies has been done in mammals (Yu et al., J. Neuroscience 2009; and J. Neurophysiol. 2013) showing that temporal plasticity can be induced in multisensory neurons with repeated visual-auditory stimulation, and that such experience enhances excitability). However, the biophysical bases for these changes, and other changes induced by experience on multisensory integration are unknown, and unlikely to be known unless approached in a more tractable system like that used by these authors.

3) It is important to return to the issue of the stimuli used in the context of development. It must be acknowledged that this was an excellent choice to drive the postsynaptic neuron – and essential in the explant. However, it does not accurately capture differences in the inputs to the tectum that are likely to exist at different ages. For example, given other literature, one would expect that naturally-generated inputs are substantially more variable (in timing, efficacy, etc.) on a trial-by-trial basis in younger animals than in older animals. Certainly that is true in the mammal and probably in *Xenopus* as well. Given that this issue is bypassed in the current paradigm, a bit more care has to be used in the interpretation of the results. This should not be interpreted as a flaw in the study. It's just an important point for the authors to deal with and the possible alternative results with nature stimuli.

Reviewer #1 (General assessment):

This is a very well done study of the development of multisensory integration in the *Xenopus* optic tectum. The authors have done an admirable job here, and have provided some interesting and insightful observations. Their use of picrotoxin was especially helpful in dealing with their supposition that inhibitory portions of the circuit are becoming more robust with development, thereby minimizing the magnitude of the multisensory response that they see earlier in life. Their comparison to data generated from mammals is particularly helpful, and demonstrate the constancy of many of the mechanisms underlying this phenomenon across species and ecological challenges. They do note, quite appropriately so, that the use of electrical stimulation (which has no clearly defined spatiotopic referent in space) may have produced differences that are not really species-specific is an important caveat. This is especially important in their studies of within-modal stimulus pairs. Two electrical stimuli, via the same electrodes engage the same afferents at different ISIs. This condition that is difficult to relate to two visual stimuli at different locations in space, and may be more akin to a single, repeated stimulus. This makes the comparison to the visual-mechanosensory stimulus a bit tricky, and may require a stronger caveat. Nevertheless, the observation is an important one, and is thought-provoking. This point is also important with regard to the proportion of sublinear and supralinear responses in the multisensory condition. Electrical stimuli by their nature produce very different discharge patterns than do environmental stimuli. The absence of similar stimulation patterns in mammalian multisensory neurons makes it difficult to compare. Nevertheless, their findings that only the multisensory condition shows developmental changes, and that it shows an increasing sensitivity to the effects of picrotoxin, clearly indicate substantial differences from the two visual stimulus conditions – and that the nature of these differences differ from the unisensory-multisensory differences noted in mammals. Some of these issues are discussed in a recent review of the development of multisensory integration (see Stein et al., Nat Rev. Neurosci., 2014).

In summary, this is an interesting and informative study that expands our understanding of the development of multisensory integration capabilities at the physiological level. The manuscript is well-written and approachable even by the non-expert. It places the finding properly within the field, and also makes clear why this system and others like it are so well-suited for exploring the synaptic mechanisms underlying this phenomenon.

*Reviewer #2 (General assessment):*

Felch, Khakhalin & Aizenman examine the mechanisms underlying multisensory integration in the developing tadpole tectum. Prior studies in mammalian superior colliculus, largely from the Stein lab, have shown that sensory experience is required for development of multisensory integrative capacity. Felch et al. provide important new information on the cellular and synaptic mechanisms contributing to the development of multisensory integration over a developmental period when tectal circuitry matures as a result of changes in intrinsic excitability, recurrent excitation and inhibition, and changes in sensory receptive fields and topographic maps. This study uses an isolated brain preparation from *Xenopus* tadpoles, which allows ready access to tectal neurons where MSI occurs and direct stimulation of visual and mechanosensory inputs in a preparation where the tectal circuit remains intact. Felch et at used cell attached recordings to determine the outcome of changing the temporal interval between visual and mechanosensory afferent stimulation to tectal neurons compared to unisensory stimulation. In contrast to unisensory paired responses they found that shorter intervals result in faster stronger MSI responses while MSI responses were inhibited with longer intervals. They found that the population of neurons from older animals had smaller responses, and that this was due partially to individual neurons having a smaller response and narrower tuning curve. The authors then test whether inhibition is constraining MSI responses and find that blocking inhibition increases MSI responses and broadens the MSI tuning curve in neurons from older animals, so they resemble the tuning curves in younger stages. Furthermore, MSI responses were largely sublinear and this sublinear response was due to inhibition. The authors present a model in which multisensory inputs recruit both recurrent excitatory and inhibitory networks, which increase responses to MSI and sharpen tuning curves and temporal responses, respectively. The study characterized MSI at earlier developmental stages than in previous studies and demonstrates the predominant role of developmental changes in inhibition in MSI responses.

Overall, this study makes a significant contribution to our understanding of MSI. The experiments are well designed, the data are clear and carefully analyzed. The authors present a model that is well supported by their data and they discuss similarities and differences between their findings and work in other systems. I have no substantial concerns.

---

## [Author Response]

Essential revisions:

1) In the first paragraph of the Results the authors refer to previous studies of ISI in mammalian systems. The scholarship here is evolving and there is a more nuanced view of these effects and the consequences of stimulus order on the resultant response (e.g., see Miller et al., J. Neuroscience 2015).

This is very interesting, thank you for bringing it to our attention. In our experiments we didn’t observe differences between the order in which each modality stimulus was presented, but we also made sure that the responses to each modality were of comparable strength, and thus balanced. It will be interesting to see in future studies whether the ‘stronger first’ rule is preserved in the tectum, and whether this rule can be disrupted by inhibitory blockade. This would be a direct test of the model put forth in Miller et al. We have added mention of this study in the Introduction, but included a longer paragraph in the Discussion (fifth paragraph).

2) In the fifth paragraph of the Discussion they make a point about the sensitivity of timing to experience in this circuit. This is an excellent point about how the circuit involved adapts to its use. This system is ideal for exploring this issue in more detail. To my knowledge only two such studies has been done in mammals (Yu et al., J. Neuroscience 2009; and J. Neurophysiol. 2013) showing that temporal plasticity can be induced in multisensory neurons with repeated visual-auditory stimulation, and that such experience enhances excitability). However, the biophysical bases for these changes, and other changes induced by experience on multisensory integration are unknown, and unlikely to be known unless approached in a more tractable system like that used by these authors.

This is something we’ve thought about a lot, and agree that the recurrent circuitry of the tectum provides an ideal substrate for this type of plasticity. We’ve already shown that network plasticity works for altering tuning to pairs of unimodal stimuli, and are currently working to apply this to a multisensory protocol. We’ve added some text linking our prior findings to work in mammalian SC:

“This type of network plasticity may also help explain observations in the mammalian superior colliculus where temporal statistics of cross-modal stimuli during development and in adulthood shape the temporal tuning of multisensory responses (Xu et al., 2012, Yu et al., 2009). Using the *Xenopus* tadpole tectum preparation it should be possible to study this type of plasticity at the cellular level.”

3) It is important to return to the issue of the stimuli used in the context of development. It must be acknowledged that this was an excellent choice to drive the postsynaptic neuron – and essential in the explant. However, it does not accurately capture differences in the inputs to the tectum that are likely to exist at different ages. For example, given other literature, one would expect that naturally-generated inputs are substantially more variable (in timing, efficacy, etc.) on a trial-by-trial basis in younger animals than in older animals. Certainly that is true in the mammal and probably in Xenopus as well. Given that this issue is bypassed in the current paradigm, a bit more care has to be used in the interpretation of the results. This should not be interpreted as a flaw in the study. It's just an important point for the authors to deal with and the possible alternative results with nature stimuli.

We agree, the stimulation paradigm used in this study allowed us to have a reduced preparation and gave us excellent temporal control. These were required to really be able to dissect the underlying circuitry and will remain important tools for figuring out rules of plasticity and integration in single neurons. However, we also recognize that the spatiotemporal complexity and variability of in vivo responses will also play an important role in how these interactions occur and will be the subject of future work. We’ve strengthened this caveat by adding text:

“One possible explanation for this discrepancy is the use of bipolar stimulating electrodes in our preparation to evoke action potentials throughout an afferent pathway. […] It will be important in future studies to compare the rules underlying multisensory integration using in vivo visual and mechanosensory stimuli.”